# Study of Pressure Distribution in Floor Tiles with Printed P(VDF:TrFE) Sensors for Smart Surface Applications

**DOI:** 10.3390/s23020603

**Published:** 2023-01-05

**Authors:** Asier Alvarez Rueda, Philipp Schäffner, Andreas Petritz, Jonas Groten, Andreas Tschepp, Frank Petersen, Martin Zirkl, Barbara Stadlober

**Affiliations:** 1Joanneum Research Forschungsgesellschaft mbH, Franz-Pichler-Straße 30, 8160 Weiz, Austria; 2Parador GmbH, Millenkamp 7-8, 48653 Coesfeld, Germany

**Keywords:** smart structure, distributed pressure sensor, flexible device, P(VDF:TrFE), finite element method

## Abstract

Pressure sensors integrated in surfaces, such as the floor, can enable movement, event, and object detection with relatively little effort and without raising privacy concerns, such as video surveillance. Usually, this requires a distributed array of sensor pixels, whose design must be optimized according to the expected use case to reduce implementation costs while providing sufficient sensitivity. In this work, we present an unobtrusive smart floor concept based on floor tiles equipped with a printed piezoelectric sensor matrix. The sensor element adds less than 130 µm in thickness to the floor tile and offers a pressure sensitivity of 36 pC/N for a 1 cm^2^ pixel size. A floor model was established to simulate how the localized pressure excitation acting on the floor spreads into the sensor layer, where the error is only 1.5%. The model is valuable for optimizing the pixel density and arrangement for event and object detection while considering the smart floor implementation in buildings. Finally, a demonstration, including wireless connection to the computer, is presented, showing the viability of the tile to detect finger touch or movement of a metallic rod.

## 1. Introduction

Current technology trends require a higher presence of pressure sensors integrated into their surroundings with the aim to increase the degree of automation or to enable condition monitoring [1,2,3]. Be it for medical applications [4], industrial environments [1], positional tracking [2,5,6,7], or other fields [8,9], pressure sensors have become ubiquitous. In smart building/home environments, pressure sensitive floor tiles open new possibilities of movement, occupancy, and event detection without the need of privacy-violating video surveillance, which is especially relevant in non-public areas, where video surveillance is forbidden. These functionalized floor tiles make up a “smart surface”, which usually also comprises a data collection and (wireless) transmission unit for analysis. For different sensor configurations and designs in smart floor applications, several models have been developed to interpret the sensor data [5,10,11,12]. However, there are hardly any systematic investigations on the relationship between the mechanical stimuli on the floor’s surface (pressure distribution) and output signal of the sensor elements. The understanding of this relationship is decisive to optimize the density and arrangement of sensor elements. On the one hand, this optimization should lead to lower costs and less materials needed for fabrication, thus saving resources, and on the other hand, it should ensure events of interest can be unambiguously resolved at the level of the sensor unit.

As for the sensor units, there are many concepts in the literature based on either capacitive [13] or resistive [14], triboelectric [15] or piezoelectric [16] sensing mechanisms.

A capacitive sensor uses changes in capacitance between two electrodes due to a change in distance between them to detect changes in the applied force. Hoffmann et al. [17] used commercial capacitive floor sensors to study gait patterns. However, as Middleton et al. [13] noted, leakiness may be a problem, and it might require a complex circuit design to avoid it.

Strain gauges based on the piezoresistive effect are a cheap and reliable alternative for strain measurement. Chen et al. [14] presented a flexible matrix of strain gauges that allowed a directional measurement of strain while conforming to the surface on which the gauge matrix was placed. According to their paper, they plan to use it on machinery or different kind of structures to detect damage. The main disadvantage of this approach is the sensor units must be externally powered since strain gauges measure strain by a change in the internal resistance.

In contrast to strain gauges, triboelectric transducer units allow for a self-powered device. This approach can be used both for sensing as well as for energy harvesting and is currently a hot research topic [14,17,18,19,20]. However, triboelectric transducers bring many challenges because the movement they require is not just a deformation but either contact-separation or sliding [21]. They also have high impedance, open circuit voltages in the hundreds of volts range, and the output is generally an alternating current (AC) requiring adapted electronics [22].

Finally, piezoelectric transducers are a more classical approach for creating self-powered sensors. In contrast to triboelectric transducers, with piezoelectric transducers even small deflections can produce sufficiently high signals with good linearity, and the energy gained can be used without complex impedance matching circuits [23].

The use of piezoceramic materials, such as PZT, in smart floor environments for sensing or energy harvesting is not a new topic [6,23,24]. In this context, piezoelectric polymers, such as polyvinylidene fluoride (PVDF) and its copolymer poly(vinylidene-trifluoroethylene) (P(VDF:TrFE)), are of special interest. In contrast to piezoelectric ceramics, the piezoelectric polymers are highly flexible, can be printed from solution on foil substrates in various shapes, and need only low processing temperatures of around 135 °C [25]. Printing techniques, such as screen printing, enable large sensor areas with complex pixel array designs [26]. These techniques make it easier to adapt the sensor layout to the particular case. At the same time, flexible devices can be conformably applied to the targeted surfaces by gluing or lamination, thus avoiding bulky mechanical constructions. Previously, we integrated P(VDF:TrFE)-based sensors in floor tiles in an augmented floor [27], whereby the sensors were reacting on the users’ stepping or jumping. The respective response signals were processed by the AI to produce an auditory augmentation of the motion. An in-depth modeling of the sensor response to pressure and force excitations was not part of this study. 

In this work, we study the integration of fully printed sensor arrays featuring PyzoFlex^®^ technology [28] into floor tiles. The integration comes without bulky mechanical constructions. The main aim is to understand how the pressure distribution on top of the floor tile induces a signal in the sensing layer, which is placed below the tile. The pressure distribution caused by the event of interest, e.g., a person stepping on the floor, spreads through the floor and causes a locally varying response in the sensor array (Figure 1). As mentioned in the beginning, understanding this relationship helps to optimize the density of sensor pixels required to detect a given event, while reducing crosstalk between pixels and lowering the implementation costs. To this end, we compare the experimental values obtained from a 3 × 5 sensor array under controlled excitation conditions with the simulation using a finite element model of the smart floor.

## 2. Materials and Methods

### 2.1. Fabrication of Smart Floor Tile

As the first step toward fabrication of the floor tile, a sensor matrix was realized on a flexible substrate (design and layer stack depicted in Figure 2) following a process reported earlier [29,30]; see Figure 2. In this case, five layers were screen-printed, where the ink and mesh types were chosen to achieve the desired layer thicknesses and homogeneity. The layers were as follows, with screen mesh types indicated in brackets:Conductive polymer layer out of poly(3,4-ethylenedioxythiophene) polystyrene sulfonate (PEDOT:PSS) (CleviosTM SV4 Stab, Heraeus Deutschland GmbH & Co. KG, Hanau, Germany) acting as the bottom electrode (100–35);

2.P(VDF:TrFE)_80:20_ ink (Piezotech^®^ FC20, Piezotech Arkema, Pierre-Benite Cedex, France) as ferroelectric material (24–150);3.Another layer of PEDOT:PSS for the top electrode (100–35);4.Connection lines with silver ink (Bectron CP6611) (59–32);5.Protection layer (UVLM 2, Marabu GmbH & Co. KG, Tamm, Germany) (150–31).

The transducer stack was printed with a Thieme LAB 1000 screen-printing machine. The inks were deposited on a polyethylene terephthalate (PET) foil substrate with 125 µm thickness (Melinex ST505, DuPont Teijin Films (Luxembourg) SA, Contern, Luxemburg). The substrate was selected for its good wetting behavior for the screen-printing technique as well as its chemical and mechanical resistance. PEDOT:PSS was chosen as the electrode material for its environmental friendliness and supreme compatibility with the substrate and P(VDF:TrFE), partly due to the fact it uses water as a solvent in contrast to many other inks (e.g., silver). It forms smooth interfaces with the ferroelectric layer, which is crucial during electrical poling, and is also mechanically more durable compared to most silver inks. All layers, including the ferroelectric layer, were cured at 120 °C. Finally, the substrate was cut manually, and the sensor unit was poled, following the procedure explained in the next section.

After fabrication of the sensor matrix, the transducer was cut and glued to the bottom of a floor tile (Modular ONE, Parador GmbH, Coesfeld, Germany) with a layer of commercial wood glue (Redocol) (see Figure 3). The device was then left to dry at room temperature for 24 h.

### 2.2. Material Characterization

Both the substrate’s and floor tile’s mechanical parameters were unknown; thus, tensile tests and three-point flexural tests were performed using an Instron 3342 universal testing machine. In both cases, the Young’s moduli were obtained from the tests and are listed in Table 1. For the PET substrate, rectangular strips of 1 × 15 cm^2^ were cut in orthogonal directions, and a tensile test was performed at 1.2 mm/s speed. For the floor tile, a three-point flexural test was preferred. A 15 × 2 cm^2^ sample of the 8 mm thick floor tile was cut out. Using item aluminum profiles, a holder with a 10 cm long gap was assembled. A T-shaped profile was tightly fixed at the tensile tester’s moving end, and then, a force of up to 90 N was applied at the center contact line to the bar while recording the displacement. Images of both testing setups are available in the Appendix A. The Young’s modulus was calculated from the resulting force-displacement curves.

Cross section views of the sensor foils were acquired using SEM imaging of microtome cuts. The samples were cooled with liquid nitrogen, and a clear cut was obtained with a microtome, exposing the layered structure of the transducer. The images of the cross section were taken with a Zeiss Gemini SEM (as part of a RAITH eLINE system).

The ferroelectric properties of the P(VDF:TrFE) layer, such as remnant polarization and the coercive field, can be extracted from the poling. During poling, a sinusoidal voltage with an amplitude up to 800 V (approximately double the coercive field of the ferroelectric layer) at 10 Hz frequency was applied to each sensor pixel followed by a positive-up-negative-down (PUND) pulse sequence while recording the poling current [32,33]. 

The apparent piezoelectric coefficient,$\;d_{33}$, was measured for each pixel before gluing the sensor sheet to the floor tile. By doing so, a pneumatic stamp applied up to 90 N over the sensor pixel following a triangular-shaped pulse at 0.5 Hz. The transducer was situated over a force sensor. To ensure the force was applied homogeneously over the pixel, a rectangular PET foil of 1 × 1 cm^2^ was put between the sensor pixel and the stamp. Both force sensor output and transducer pixel output were measured simultaneously with a National Instruments PXI card. The current signal from the transducer was converted to a voltage signal using a transimpedance amplifier (TIA) and integrated to obtain the piezoelectric charge. The apparent $d_{33}$ parameter was then determined from a linear fit of the charge-force plot. An example of one measurement as well as an image of the setup is shown in the Appendix A. 

### 2.3. Floor Testing

We tested the smart floor tile in a realistic use case scenario with the setup shown in Figure 3. It comprises (from top to bottom) a metallic stamp head mounted to the moving end of the tensile tester, the smart floor tile, an underlayment layer (Easycut^®^), and a rigid base. The stamp head consists of two parts specifically designed for these experiments: an upper part screwed to the load cell of the tensile tester and a lower tiltable part with a flat face that is attached via a magnetic sphere to the upper part. This way, shear forces during compression are reduced to a minimum while the pressure is distributed homogeneously over the circular stamp face. To match the size of the matrix pixels, we used a 1 cm^2^ circular stamp. The stamp’s radius is then 0.56 cm. We applied a linearly varying force over the floor ranging from 0 to 90 N. The signal of each pixel was read out using a Keithley 6517A electrometer configured for charge measurement and recorded using a Dewesoft Krypton data acquisition (DAQ) system. We first tested the output at the central pixel with two movement speeds (10 and 150 µm/s) to ensure the independence of the signal output from the movement speed. Then, we measured the output of the whole matrix by measuring the response of each pixel at a time under the same excitation conditions, i.e., without changing the stamp position with respect to the smart floor tile.

### 2.4. Finite Element Model

The model, as summarized in Figure 4, was implemented using COMSOL Multiphysics^®^ 6.0. The model’s feature sizes span from the micrometer thick polymer layers of the sensor matrix to the 8 mm thick floor tile. To improve the solution time as well as design flexibility, the system was simplified into a layered block consisting of the floor tile, sensor substrate, and the piezoelectric layer. The mechanical response of the system will be dominated by the floor’s parameters since its thickness is an order of magnitude greater than the other elements. Because of this, the protection layer is not included in the model, and electrode layers are only considered as boundary conditions for the electrostatic equations. The thin adhesive layer is not considered either, as apart from the edge regions of the sensor substrate, it transfers the strain present at the floor bottom directly into the substrate with negligible losses.

To mimic the contact pressure exerted by the stamp, a constant boundary pressure is applied over a circular area with radius, *a*, given by
(1)$$p_{0}=\frac{F_{N}}{\pi a^{2}},$$
where $F_{N}\;$ is the normal force applied by the cylindrical head, and *a* is equal to 0.56 cm in corresponding to the dimension of the experimental stamp.

The floor tile is modeled as a linear elastic isotropic material. The isotropic symmetry is justified by the results of the material tests. For this application, the involved strains are expected to be small ($\left|\epsilon \right|<10^{-3}$), and thus, a linear model should hold in spite of the possible non-linearities coming from the floor tile’s and the polymers’ (P(VDF:TrFE), PET) compositions. The mechanical and electrical material parameters used in the FEM model are summarized in Table 1.

For the piezoelectric layer and the substrate, we chose an orthotropic linear elastic model. To describe the piezoelectric coupling, a dimensional model was implemented, where the poling direction is defined to be the 3 direction [34]. Accordingly, the piezoelectric coefficient matrix *d* can be expressed via the remnant polarization, *P_r_,* and the compliance matrix *s* as [35]:(2)$$d_{ij}=-P_{r}\left(\begin{array}{cccccc} 0 & 0 & 0 & 0 & 0 & 0\\ 0 & 0 & 0 & 0 & 0 & 0\\ s_{31} & s_{32} & s_{33} & 0 & 0 & 0 \end{array}\right),$$
where $s_{31}=s_{32}=-\nu /Y_{l}$ and $s_{33}=-\nu /Y_{t}$ with ν being the Poisson’s ratio. The sensor charge response is measured in short-circuit mode (zero voltage, i.e., zero electric field) using a TIA. The short-circuit condition is applied to the whole top boundary of the piezoelectric layer. The sensor array is then implemented as a set of areas, *A_i_*, representing the individual pixel electrode areas. The charge collected by the *i*-th pixel, *Q_i_*, is calculated from the dielectric displacement *D* as:(3)$$Q_{i}=\iint_{A_{i}}D_{3}\left(x,y,z\right)\;dA,$$
where the polarization direction of the dielectric film is the 3-direction, and *D*_3_ is given by the piezoelectric constitutive equations at the absence of an electric field. In the finite element model the electrodes are not considered as separate domains but appear as the integration domains, *A_i_*, of Equation (3).

Finally, the boundary between the bottom layer and the surroundings is modeled as a Winkler foundation [36]. This reduces the problem to the determination of a single parameter, *K_eff_*, where the best match to the experiment was found for 150 N/cm^3^. The boundary is fixed laterally with a small restoring force to compensate spurious lateral movements while avoiding clamping the strain in the lateral direction. 

A quadrilateral mesh with a maximum element size of 1.5 mm and automatic refinement was used, and the results were obtained with a stationary solver (see Appendix A). 

### 2.5. Data Acquisition Unit

A development kit (Joanneum Research, Graz, Austria) was used as a data logger to record the charge responses of the smart floor tile sensor matrix during demonstration. It was comprised of a Raspberry Pi 3B+ carrying a self-developed hock-up board containing the analog input stages and a multichannel analog-to-digital converter (ADC). The ADC samples with up to 1.4 kSps/channel at a 24-bit vertical resolution. The Raspberry PI communicates with the ADC on the hock-up board via a serial peripheral interface (SPI). A total of 16 optional sensor signals can be read upon amplification by an operational amplifier circuit (JFET-based). To this end, a charge amplifier circuit (charge-to-voltage converter) with a feedback capacity of *C* = 1 *nF* was chosen. The system was powered by a battery pack (5 V supply voltage) with a charging capacity of 5000 mAh.

## 3. Results and Discussion

### 3.1. Mechanical and Electrical Properties

Figure 5a displays SEM images of the cross section of a representative pixel element with the screen-printed layer stack. The actual transducer has a thickness of 4.4 µm; it comprises a 0.3 µm thin top and bottom electrodes made of PEDOT:PSS and the piezoelectric layer made of P(VDF:TrFE) with 3.8 µm thickness. The thin-layer thicknesses were achieved by using adequate mesh types based on the individual layers’ ink viscosities; see Section 2.1. The lamellar structures typical for the ferroelectric β-phase P(VDF:TrFE) are clearly visible [37]. 

From the poling current and the voltage, the dielectric displacement was derived as a function of the electric field. Using the PUND method [33] allowed to eliminate the dielectric response and obtain the “pure” hysteresis plot of the spontaneous polarization, *P,* vs. electric field, *E*, as shown in Figure 5b for a poling frequency of 0.5 Hz. The remnant polarization, $P_{r}$, was 81 mC/m^2^, and the coercive field, $E_{c}$, was ~62 MV/m, typical for the P(VDF:TrFE) family [34].

To build a realistic model for the FEM simulation, the mechanical properties of the floor tile and sensor substrate were measured. Both materials showed an almost linear behavior at low strain in the tensile experiments. In the case of the floor tile, it was still linear up to 0.5%, while PET showed slight deviations starting from 0.8 % (see Appendix A). As indicated in Table 1, a Young’s modulus of 2.6 ± 0.1 GPa was derived for the floor tile. For the PET substrate, the moduli were 4.8 ± 0.2 GPa for the longitudinal and 4.0 ± 0.2 GPa for the transversal direction. In both cases, the samples were cut in orthogonal directions to test the assumption of isotropic symmetry. In the case of the floor tile, no difference between the two cutting directions was found. For the PET layer, as already indicated, a deviation of 0.8 GPa was found in the Young’s modulus between the two orthogonal directions. It was not possible to measure the Poisson’s ratio by this method; thus, we took a literature value of 0.4 for the PET [31] and used the same value for the floor tile, whereas for the P(VDF:TrFE) we found a value of 0.3 in a previous work [32].

Finally, as revealed in Figure 5c, the average measured apparent piezoelectric coefficient, $d_{33}$, was −26.7 ± 0.7 pC/N. Equation (2), which links the piezoelectric parameter with the remnant polarization, predicts an intrinsic $d_{33}^{i}\;$ value of −37 pC/N for the measured remnant polarization of about 81 mC/m^2^. This discrepancy can be understood by considering that the piezoelectric layer is not free to expand laterally but is constrained and clamped by the presence of the substrate and by friction. Accordingly, a lateral stress builds up, meaning the measured piezoelectric *d*_33_ parameter cannot correspond to the intrinsic value; the measured *d*_33_ is therefore denoted as “apparent” $d_{33}$. If we consider the clamping is perfect, then the intrinsic and the apparent piezoelectric parameter are related through the Poisson’s ratio of P(VDF:TrFE), ν, as [32]:(4)$$d_{33}^{i}=d_{33}\;{\left(1-\frac{2\nu^{2}}{1-\nu }\right)}^{-1}.$$

For a Poisson’s ratio of 0.3, as the one used for the FEM model, the intrinsic $d_{33}^{i}$ calculated via Equation (4) is $d_{33}^{i}$ = −35.6 pC/N, which is very close to the value of −37 pC/N predicted by Equation (2). It should be noted that according to Equation (2), the piezoelectric coupling is directly proportional to the remnant polarization, *P_r_*, which is obtained during fabrication for every produced sensor sheet, separately for every sensor pixel. Thus, the value of *P_r_* could serve as a calibration factor in the sensor signal evaluation.

### 3.2. Smart Floor Testing and Modeling

To better understand the temporal and spatial sensor response of an integrated sensor matrix, we tested its response, as indicated in Section 2.2, and compared the generated charge to the model we described in Section 2.4. Figure 6 shows the temporal response of the pixel right underneath the pressure stamp for two different excitation speeds (10 and 150 µm/s). In both cases, the drifting current was considered by removing a linear baseline from the charge time series. The charge response followed closely the force profile and varied almost linearly with the applied force, as shown in the bottom graphs of the figure. A linear fit revealed the same slope value of 36 pC/N for both excitation speeds. The simulated model, which is based on a stationary formulation, predicts a linear response with only a slightly higher slope of 37 pC/N. This was achieved, as indicated before, with an underlayment effective stiffness of 150 N/cm^3^.

To study the distribution of the pressure through the floor tile into the sensor foil, we measured the responses of all individual sensor pixel responses sequentially without moving stamp and tile. The resulting distribution over the sensor matrix is shown in Figure 7a. Even though the stamp size was similar to that of a single pixel, the charge response was not strongly concentrated over the central pixel but rather spread over the whole matrix. At the most distant outer pixels at (*X*,*Y*) = ±(1.5, 3) cm, the response had still ~1/6.6 of the magnitude of the central pixel. Quantifying this lateral spread is important to determine the minimum density of sensor pixels required to resolve specific mechanical stimuli of interest while reducing the crosstalk between pixels. The FEM model could reproduce the value at the central pixel with an error of 1.5% (see Figure 7b). The crosstalk was underestimated in this model, though with a far higher error in the signal of the closest neighbors.

To analyze how the choice of the underlayment’s parameter, *K_eff_*, affects the lateral distribution of charge response in the model, we varied the value of this parameter and evaluated the piezoelectric polarization in the continuous piezoelectric layer. As shown in Figure 8, lower values of stiffness resulted in higher polarization values over wider areas. On the other hand, a stiffer underlayment concentrated the polarization under the compressed area, and thus, crosstalk was reduced. Still, it was not possible to completely fit the simulated values to the experimental ones. Changing the lateral clamping conditions did not change this fact either. A stronger restoring force applied laterally only increased the stiffness of the boundary, resulting in a reduced spread of the polarization. This might point out the limitation of a single parameter modeling of underlayment.

In addition, the FEM model allows for a more detailed study of how the charges are generated. According to the piezoelectric parameter tensor (Equation (2)), our piezoelectric material will polarize due to stress in the 1-, 2-, or the 3-direction. As for the direction convention of Figure 4, the former two cases appear for a bending of the sensor matrix with *d*_31_ = *d*_32_; the latter case corresponds to a compression of the matrix pixels, and the corresponding piezoelectric parameter is *d_33_*. While it is not possible to select one mode or the other without changing the clamping conditions in the experiment, it is very well possible in the model to fix one of the piezoelectric coefficients and investigate its effect on the final result.

To test the claim that the main contribution to the piezoelectric response comes from a lateral bending stress in the piezoelectric layer rather than from vertical compression, we set either the *d*_33_ or *d*_31_ and *d*_32_ components of the piezoelectric parameter tensor to zero in our FEM model. By doing so, we found when both the 31- and 32-components are zero, only 1.1% of the original output signal was achieved. The major contribution thus came from the stress in the lateral 1 and 2 directions (see coordinate axes in Figure 4), which is linked by the *d_31_* component, rather than from the compressive stress in the *3*-direction. At the same time, if the displacement was set to zero at the lower boundary of the simulation domain to suppress bending of the piezoelectric layer, the response distribution was much more confined. As shown in Figure 8, this confinement of the signal decreased by reducing the boundary stiffness parametrized by $K_{eff}$ and thus making bending easier. Accordingly, most of the signal was related to a bending of the piezoelectric layer and not to the vertical compression as one might assume.

This is an important finding that should be considered when it comes to integrating such a sensor foil into a floor tile. When the active layer is positioned such that the bending of the piezoelectric layer is maximized, the absolute charge response can be clearly increased, and a large lateral spread is achieved. This way, the density of pixel elements can be reduced to detect a specific impact event. However, when a high spatial resolution is needed, the larger spread in the charge response would be detrimental. All in all, the chosen layer setup depends on the use case.

As to the long-term stability, the sensor response did not change significantly after repeated measurements. Previous tests showed the underlying PyzoFlex^®^ technology shows supreme durability and can withstand even harsh environmental changes [38].

### 3.3. Smart Floor Demonstration

Finally, as a demonstration, we constructed a wireless smart tile matrix, the setup of which is shown in Figure 9. Here, the smart tile was connected to an in-house developed 16-channel wireless data acquisition unit of which 15 channels were used for this demonstration; see Figure 10 and Section 2.5 for details. The central processing unit is based on a Raspberry PI 3B+. The unit is powered by a battery and sends the acquired data wirelessly to a computer, where the sensor matrix is reproduced in a graphical user interface (GUI). The measurements are performed at 1.4 kSps/channel. This allows simultaneous measurement of the whole matrix, albeit with lower resolution than with the electrometer. The self-charging of the amplifier’s capacitor also limits the operation time, requiring a regular discharge of the capacitor. The tile is placed over the soft underlayment layer, which amplifies the bending of the active layer in the sensor foil. This underlayment layer was the same used in the previous experiment.

We tested our smart floor device under some simple use cases. In Figure 11, we showed the smart floor device was able to detect the movement of a rolling object (full movements can be appreciated in the Appendix A). The force was applied on the top, middle, and bottom rows of the matrix. The pressure center could be clearly seen in the response, as shown in Figure 11 and Appendix A. However, even for a small, pressurized area, as the one produced by a roller, a crosstalk appeared in the nearest neighboring pixels. When the cylinder is rolled over the floor crossing different rows, the movement is reflected in the response with the three elements in a row more or less equally excited. A similar result can be seen when we touch the smart floor with the finger or the hand (Appendix A). Rapid impacts, such as by a falling object, could also be detected by the tile. This was obvious in Appendix A, where a small peak, similar to a spike, could be distinguished in the sensor’s output. In principle, the smart floor tile was sensitive enough to detect this impact, but a greater sampling rate would be needed to improve the temporal resolution and capture the dynamics of the impact.

We would like to emphasize the data acquisition unit used for the demonstration was not optimized for a permanent smart floor implementation in terms of energy consumption and supply conditions. The power consumption of the electronics during continuous readout of all 15 sensor pixels was 4.0 W at maximum sampling rate (1400 Sps/channel) and 3.3 W at lowest sampling rate (60 Sps/channel) with a 5 V supply voltage. However, the complexity and power demand of the data acquisition unit can be reduced drastically when transmitting signals only upon event detection, e.g., by sending a beacon signal when a person steps on the floor. The electronics can easily be adapted for supply with a 120/240 V power grid using a suitable power converter and regulator (e.g., StromPi 3 by Joy-IT for the Raspberry PI system).

## 4. Conclusions

The presented smart floor based on an integrated piezoelectric sensor matrix foil was capable of spatially resolving the pressure distribution applied to the floor. A finite element model of the smart floor could be established that matched the experimental data in terms of electric signal output within a maximum error of only 1.5%. The simulation revealed the piezoelectric response is 98.9% related to the bending of the floor tile ($d_{31}$ mode) rather than to the vertical pressure ($d_{33}$ mode). Therefore, the floor underlayment, more specifically, its effective stiffness, K*_eff_*, strongly affects the spread of the pressure distribution from the topside into the sensor layer. From the obtained experimental data and FEM model, the density of the sensor elements can be optimized in view of the event type to be detected while reducing the crosstalk between sensor elements and costs in implementation of the smart floor. This will accelerate the design and prototyping of smart floors based on thin piezoelectric sensing foils, such as the PyzoFlex^®^ technology, in realistic use cases.

Finally, a complete demonstration with simultaneous readout and wireless transmission of all 3 × 5 pixel elements was presented, where the individual pressure levels were visualized in a GUI. Single touch and sweeping movements were demonstrated as well as shock events as caused by falling objects. The sensitivity was sufficient to resolve even weak actuations, such as by rolling metal objects.

Future work should consider the large-scale implementation of the smart floor concept to monitor human walking and dynamic events, such as jumping, where the presented model can be used to optimize the sensor pixel density and size. Our technology may also be extended to be incorporated into other surfaces for object and movement detection, such as for smart shelfs in home or industry applications.

## Figures and Tables

**Figure 1 sensors-23-00603-f001:**
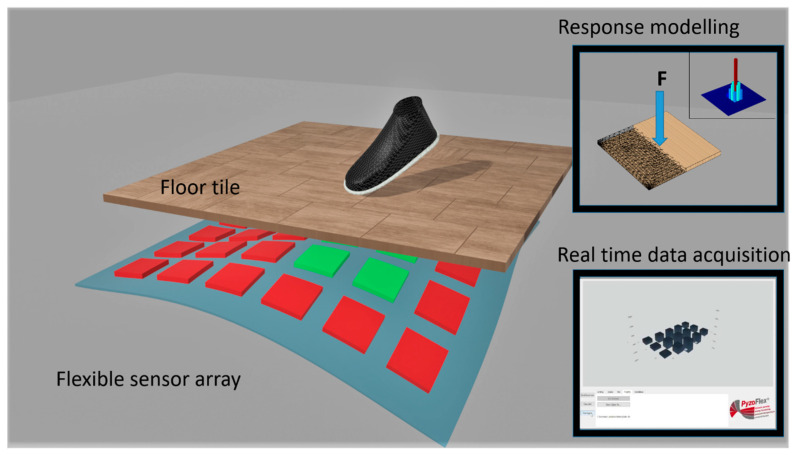
General scheme of the smart floor system. A flexible sensor matrix is attached to the bottom of a floor tile, allowing real-time position sensing of steps or any kind of pressure applied. The generated data are sent (wirelessly) to a computer for signal analysis. This data allows the development of a digital model of the system that can be used to further optimize the sensor array to the specific requirements (e.g., sensitivity, resolution) of the target application, thus avoiding unnecessary prototyping and saving costs and time.

**Figure 2 sensors-23-00603-f002:**
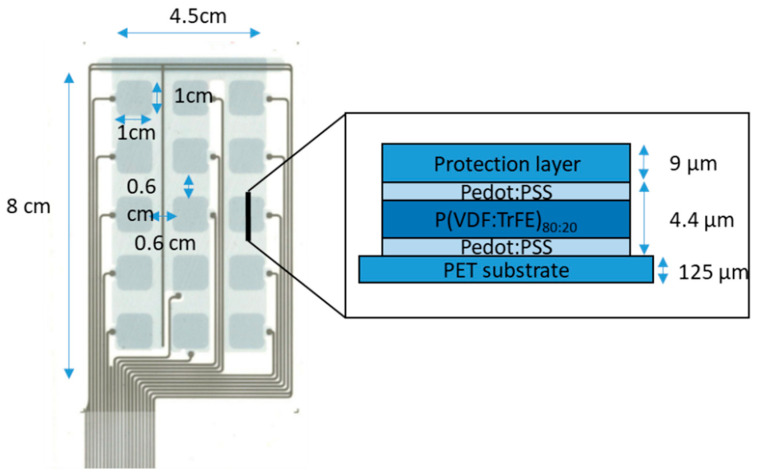
Top view of the sensor matrix and schematic of the layer stacking in one of the pixels (the layers are not in real scale). The printed device consists of a common bottom electrode for all pixels out of PEDOT:PSS, a continuous layer of P(VDF:TrFE)_80:20_ (not visible on the image) and 15 top electrodes out of PEDOT:PSS, corresponding to the 5 × 3 sensor pixel elements. The top electrodes are squares with 1 cm side length and 0.6 cm gaps between them. Every top electrode corresponding to a single pixel is independently connected via printed silver lines. The active area of the matrix is then 4.5 cm by 8 cm. The whole device, including substrate and protection layer, is around 140 µm thick.

**Figure 3 sensors-23-00603-f003:**
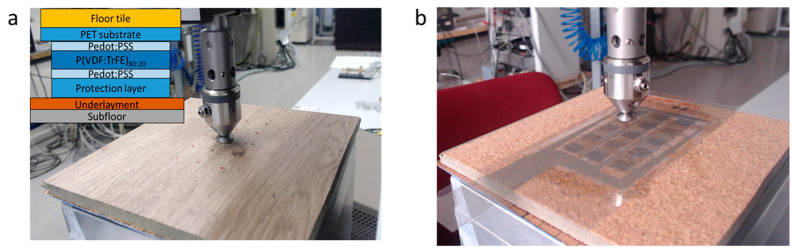
Experimental setup for testing the smart floor tile: (**a**) upper side and (**b**) flipped side. The pressure stamp applies up to 90 N with a 1 cm^2^ circular stamp head. The red dots in (**a**) are visual guides for performing the experiment and indicate the center points of each pixel. The scheme in the insert shows the layer stack of the whole setup. The floor tile is 8 mm thick. (**b**) Printed sensor matrix glued to the bottom side of the floor tile.

**Figure 4 sensors-23-00603-f004:**
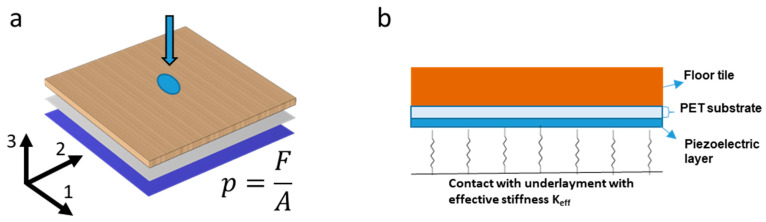
FEM model with (**a**) 3D scheme and (**b**) description of the individual domains. The contact pressure is applied as a boundary condition on top. The sensor pixel array is represented by a set of electrode areas on the top boundary of a continuous piezoelectric layer. This simplifies the study of the optimal distribution of the pixels as well as the possible crosstalk between them. The underlayment is modeled as a spring support with an effective stiffness *K_eff_*.

**Figure 5 sensors-23-00603-f005:**
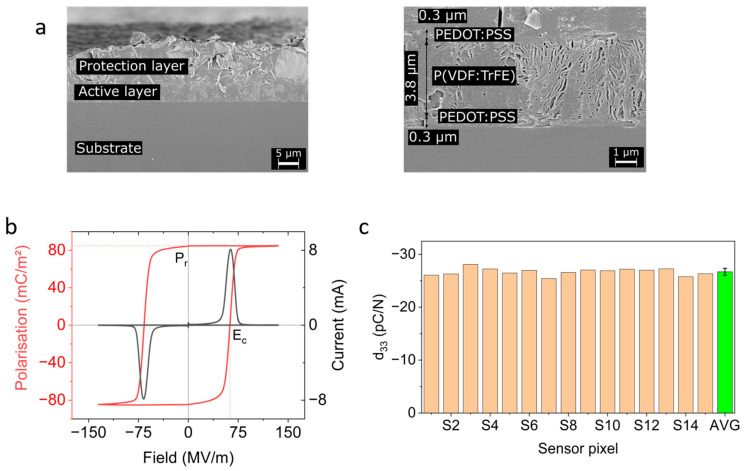
Characterization of the piezoelectric sensor pixels. (**a**) SEM image of the layer stack and enlarged view of the active layer comprising top and bottom electrodes and the ferroelectric P(VDF:TrFE) layer. The lamellar structure of P(VDF:TrFE) can be seen clearly. (**b**) Polarization (red line) and poling current (black line) of a single pixel, the composition of P(VDF:TrFE) was 80:20. Here, only a single period is shown. The polarization curve was obtained with the PUND method, and thus, the dielectric response is removed. Remnant polarization, *P_r_*_,_ and coercive electric field, *E_c_*_,_ are also indicated. The remnant polarization is the polarization at zero field and as indicated in the text, is related to the strength of piezoelectric response. The coercive field is the field at which the polarization changes sign. The poling procedure is performed up to twice the coercive field, *E_c_,* to ensure saturation of the polarization. (**c**) Apparent *d*_33_ piezoelectric parameter for each pixel (“S1” through “S15”) and average value of the whole device (“AVG”). The error bar indicates the standard deviation of the average.

**Figure 6 sensors-23-00603-f006:**
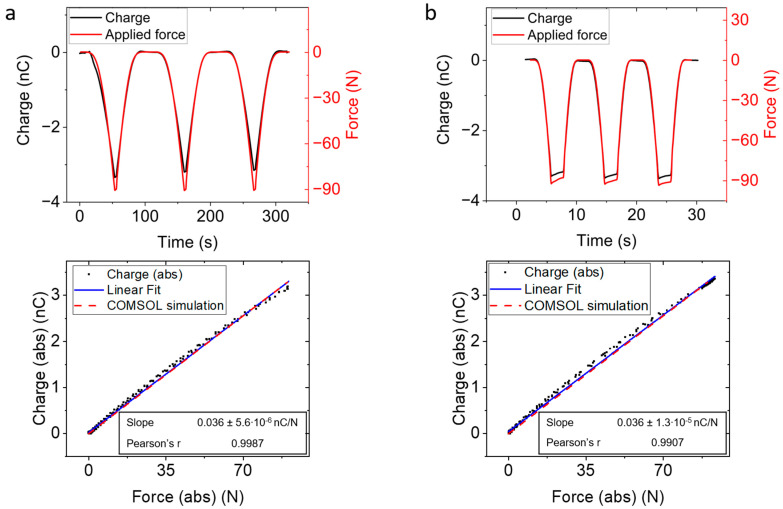
Temporal response of the central pixel for movement speed of (**a**) 10 µm/s and (**b**) 150 µm/s. On top, force and collected charge are plotted against time. On bottom, charge vs force is plotted, also including a linear fit to the experimental values (red dashed line). In blue, the simulated charge output is shown. We see both lines are close but start deviating as the force is increased. In both cases, the simulated output overestimates the experimental value. At the same time, we observe the charge output is the same for both velocities.

**Figure 7 sensors-23-00603-f007:**
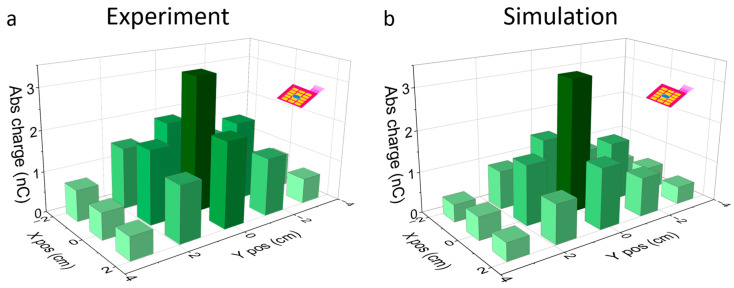
(**a**) Output of a sensor matrix when excited by a 1 cm^2^ stamp applying 90 N at 50 µm/s. (**b**) Simulated sensor matrix output for a 90 N load applied on top of the floor tile domain spread in a circular 1 cm^2^ area. In both cases the crosstalk is present, although its magnitude is underestimated in the simulation. For this FEM simulation, we used a $K_{eff}$ value of 150 N/cm^3^. This value showed the best agreement for the central pixel.

**Figure 8 sensors-23-00603-f008:**
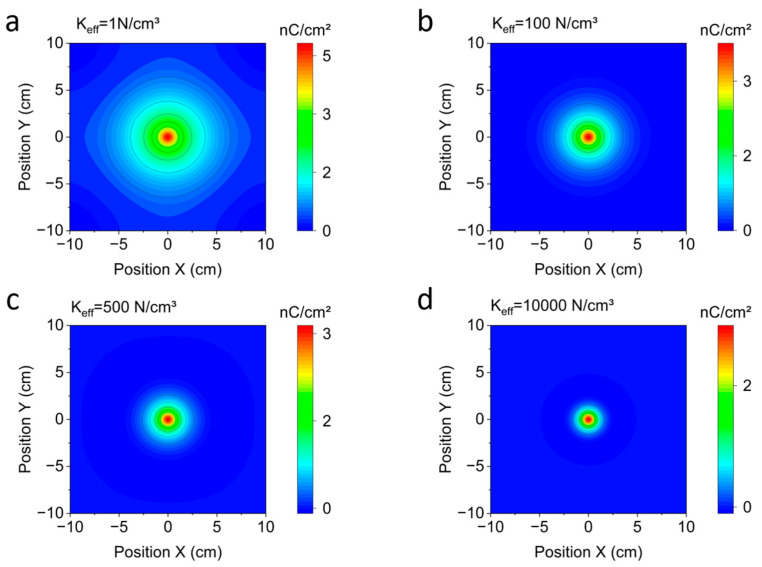
Stress-induced polarization (z-component) in the piezoelectric layer for different effective stiffness values of the underlayment layer, with *K_eff_* of (**a**) 1 N/cm^3^, (**b**) 100 N/cm^3^, (**c**) 500 N/cm^3^, and (**d**) 10,000 N/cm^3^. In all cases, a force of 90 N is applied on the top boundary of the floor tile layer. As *K_eff_* increases, the area with nonzero polarization is reduced, concentrating in the area right underneath the stamp, and the maximum polarization is also reduced. The color maps are scaled to the maximum values in the individual graphs.

**Figure 9 sensors-23-00603-f009:**
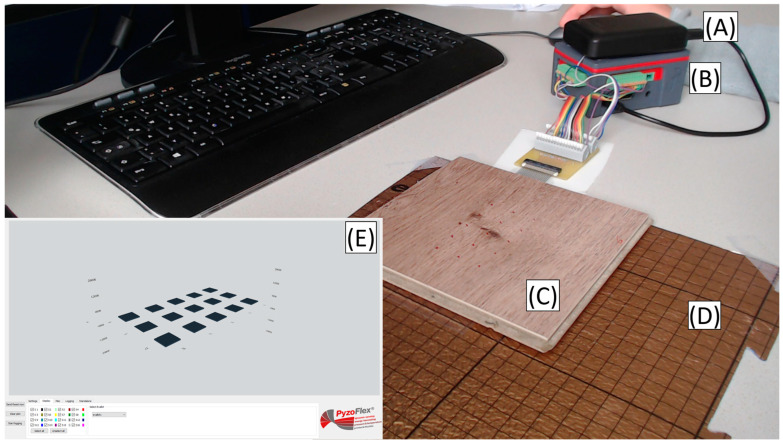
Demonstration setup, consisting of: (**A**) battery, (**B**) 16-channel wireless data acquisition unit, (**C**) smart floor tile, (**D**) underlayment layer, and (**E**) GUI on the PC.

**Figure 10 sensors-23-00603-f010:**
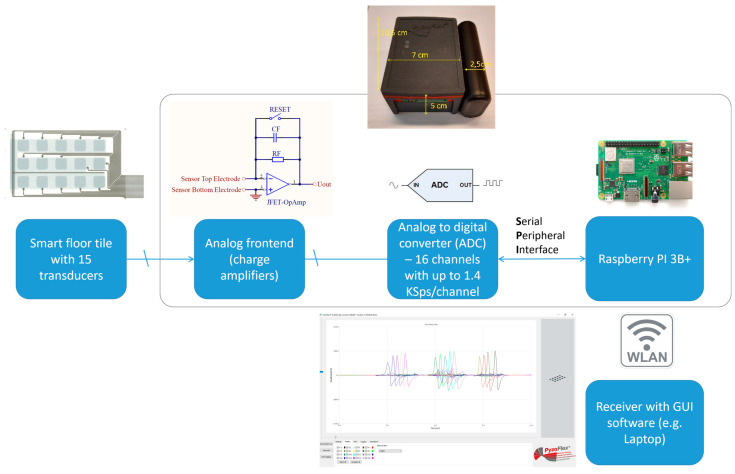
Block diagram of the data acquisition unit in combination with the smart floor tile matrix. The charge signals from each sensor pixel are amplified and converted with an ADC. The logged sensor data is then continuously transferred wirelessly to a monitoring station for display and post-processing. The electronics fits in a housing with 10.5 × 7 × 5 cm^3^ and is powered by a battery pack. See Section 2.5 for further details.

**Figure 11 sensors-23-00603-f011:**
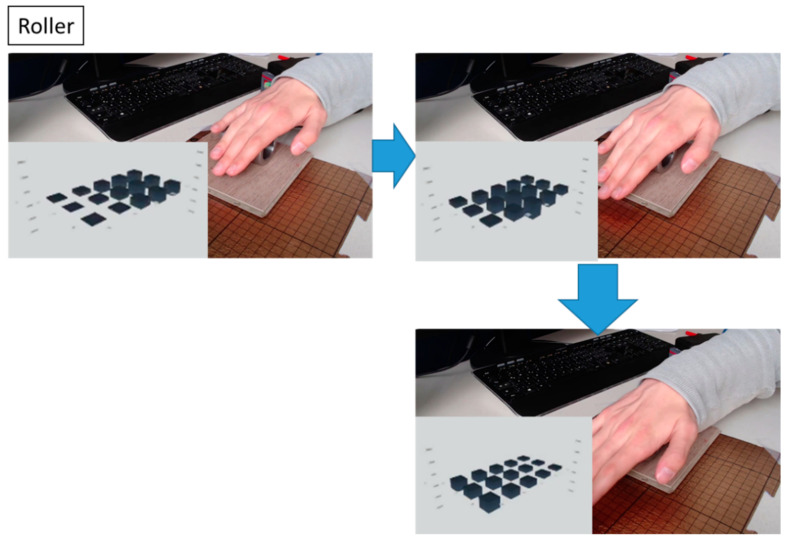
Testing of the sensor matrix with a roller moving over the matrix using the demonstration setup. The insets show the individual pixel responses, as visualized in the GUI. The matrix can clearly capture the movement of the roller. It also shows how the localized pressure applied by the roller on top of the floor spreads laterally in the sensor layer, thus creating a crosstalk between pixels. In this situation, a less dense matrix design could achieve a similar resolution with a reduced material effort and thus costs of fabrication. The established smart floor model helps optimize the pixel density for the individual use cases. The full video is available in the Appendix A together with other tested scenarios.

**Table 1 sensors-23-00603-t001:** Mechanical material parameters of the layers. Poisson’s ratios were taken either from literature (for PET) or estimated based on FEM simulations. The Young’s moduli were obtained by tensile tests (PET), three-point flexural test (for the floor tile) or from previous works (piezoelectric layer based on P(VDF:TrFE)).

Parameter	Floor Tile	PET Substrate	Piezoelectric Layer P(VDF:TrFE)
Thickness *t*	8 mm	175 µm	3.8 µm
Young’s modulus *Y*	2.6 GPa (isotropic)	*Y_l_* = 4.8 GPa; *Y_t_* = 4.0 GPa (orthotropic)	*Y_l_* = 1 GPa; *Y_t_* = 2.2 GPa (orthotropic)
Poisson’s ratio *ν*	0.4	0.4 [31]	0.3

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
