# Peer review of "Study of Pressure Distribution in Floor Tiles with Printed P(VDF:TrFE) Sensors for Smart Surface Applications"

_sensors, 2023, doi:10.3390/s23020603_

Round 1
Reviewer 1 Report
The authors describe the integration of all-plastic PVDF-PEDOT piezoelectric sensors into floor tiles to create a pressure sensing system for smart surface applications. Using both experimental setups and simulation they finally demonstrate a usable large-scale model which can be powered by battery. This is a rare demonstration of a large system for such sensors, and I think this paper is worth publishing with some minor corrections and additions.
1) The authors should include in their methods how they built the smart floor demonstration. Details such as the battery and the circuit diagrams will be useful. I do not have access to the supporting information. If that is described there, I would prefer it in the main manuscript.
2) Power considerations are important for transitions into viable technologies. The authors should provide estimates of the power consumed and whether it can be compatible with existing power standards, i.e. it can run on a 120V/240V grid
3) Please use 'demonstration' instead of 'demonstrator'.
4) Line 30: Be it for medical applications, industrial environments, positional tracking, or other fields, pressure sensors ....
5) Line 250: 'functikon' should be 'function'
6) Line 293: "modelization" should be 'modelling'
7) Please increase the thickness of the lines in your graphs, they are barely visible when they are printed out.
Reviewer 2 Report
In this work, the authors present a smart, fully printed pressure P(VDF-TrFE) sensor array in floor tiles. A finite element floor model was established to understand how the pressure distribution on top of the floor tile induces a signal in the sensing layer. The proposed model will be helpful to optimize the pixel density and sensor arrangement, and also can help to reduce crosstalk between pixels and lower the implementation costs. A demonstrator in this work exhibited a wireless connection from the sensors array to the computer, which shows its high potential in practical application. The results are novel and interesting. The paper is well organized. The following issues should be clearly addressed before it can be accepted.
1. In too many places (about 6 places), ref no. 7 was used. A review paper can contain too much complex information. The author should find specific references to replace it in a suitable place.
2. On page 4, line 122, could please give a short introduction about Melinex ST505 substrate? Which company it was produced from, and why did the author choose it?
3. On page 4, line 124, which temperature was used for piezoceramic processing? The exact temperature range should be given rather than a reference article.
4. On page 7, the last paragraph, the author presents the young’s modulus. Why young’s modulus is measured for this paper? What’s the meaning of the measurement of young’s modulus?
5. On page 8, Figure 5 a (right image), the author used different colours to mark the different layers. It is hard to observe the true layers in the image. Could the author give an SEM image without the colourful filter?
6. On page 8, Eq. (4), the explanation for ? is missing. Does it present the Poisson ratio value?
7. On page 8, line 278 and Table 1. Why the Poisson ratio value of PET and floor tile was proposed? They are not used in the later description.
8. Until now, the area of smart floor examples in the paper is very small. Could the author please give a floor that is big enough for human walking or jumping to exhibit the output of the sensor matrix?
9. In the review system, I cannot find any supporting information, did the author supply the supporting materials?
Reviewer 3 Report
Dear authors,
please find in the attachment all the comments.
Good luck

Round 2
Reviewer 2 Report
No more comments.